# mRNA BNT162b Vaccine Elicited Higher Antibody and CD4^+^ T-Cell Responses than Patients with Mild COVID-19

**DOI:** 10.3390/microorganisms10061250

**Published:** 2022-06-18

**Authors:** Federica Zavaglio, Irene Cassaniti, Josè Camilla Sammartino, Stelvio Tonello, Pier Paolo Sainaghi, Viola Novelli, Federica Meloni, Daniele Lilleri, Fausto Baldanti

**Affiliations:** 1Microbiology and Virology Department, Fondazione IRCCS Policlinico San Matteo, 27100 Pavia, Italy; fede.zavaglio90@gmail.com (F.Z.); i.cassaniti@smatteo.pv.it (I.C.); jose.sammartino@iusspavia.it (J.C.S.); f.baldanti@smatteo.pv.it (F.B.); 2Immunoreumatology Laboratory, Center for Translational Research on Autoimmune and Allergic Disease-CAAD, University of Piemonte Orientale, 28100 Novara, Italy; stelvio.tonello@med.uniupo.it (S.T.); pierpaolo.sainaghi@med.uniupo.it (P.P.S.); 3Internal Medicine Laboratory, Department of Translational Medicine, University of Piemonte Orientale, 28100 Novara, Italy; 4Immunorheumatology Unit, Division of Internal Medicine, “Maggiore della Carità” Univerisity Hospital, 28100 Novara, Italy; 5Medical Direction, Fondazione IRCCS Policlinico San Matteo, 27100 Pavia, Italy; v.novelli@smatteo.pv.it; 6Research Laboratory of Lung Diseases, Section of Cell Biology, Fondazione IRCCS Policlinico San Matteo, 27100 Pavia, Italy; f.meloni@smatteo.pv.it; 7Department of Clinical, Surgical, Diagnostic and Pediatric Sciences, University of Pavia, 27100 Pavia, Italy

**Keywords:** SARS-CoV-2, mRNA BNT162b vaccine, antibody response, T-cell response, cytokine production

## Abstract

We compared the development and persistence of antibody and T-cell responses elicited by the mRNA BNT162b2 vaccine or SARS-CoV-2 infection. We analysed 37 post-COVID-19 patients (15 with pneumonia and 22 with mild symptoms) and 20 vaccinated subjects. Anti-Spike IgG and neutralising antibodies were higher in vaccinated subjects and in patients with pneumonia than in patients with mild COVID-19, and persisted at higher levels in patients with pneumonia while declining in vaccinated subjects. However, the booster dose restored the initial antibody levels. The proliferative CD4^+^ T-cell response was similar in vaccinated subjects and patients with pneumonia, but was lower in mild COVID-19 patients and persisted in both vaccinated subjects and post-COVID patients. Instead, the proliferative CD8^+^ T-cell response was lower in vaccinated subjects than in patients with pneumonia, decreased six months after vaccination, and was not restored after the booster dose. The cytokine profile was mainly T_H_1 in both vaccinated subjects and post-COVID-19 patients. The mRNA BNT162b2 vaccine elicited higher levels of antibody and CD4^+^ T-cell responses than those observed in mild COVID-19 patients. While the antibody response declined after six months and required a booster dose to be restored at the initial levels, the proliferative CD4^+^ T-cell response persisted over time.

## 1. Introduction

Immunization against SARS-CoV-2, the causative agent of a global outbreak of a respiratory tract disease referred to as COVID-19, occurs via natural infection or vaccination. Several studies reported that higher titres of IgG and neutralising antibodies (Nt Ab) were detected in patients with severe SARS-CoV-2 infection compared to patients with mild symptoms [1,2,3,4,5,6,7]. SARS-CoV-2 spike (S) protein reactive T-cells were identified in immunocompetent (IC) patients suffering from moderate, severe, and critical COVID-19 [8], and a dominance of CD4^+^ T-cell over CD8^+^ T-cell response was observed in severe and mild COVID-19 patients [9]. In addition, patients with pneumonia developed higher memory CD4^+^ and CD8^+^ T-cell responses than patients with mild symptoms [3]. Immunological memory could persist for at least 15 months after infection [3,10,11,12,13], and may protect the IC population from SARS-CoV-2 secondary infections [14]. The mRNA vaccine BNT162b2 [15] was approved after showing safety and a 95% efficacy in preventing COVID-19 [16]. The B- and T-cell responses elicited by the vaccine have a crucial role in the protection from SARS-CoV-2 infection and disease and in the establishment of a long-term memory response. Vaccinated subjects developed Nt Ab and specific T-cell responses after two doses [17], and several studies analysed the persistence of immune response until 6–8 months post-vaccination [18,19,20,21,22]. The immune response post-vaccination and post-infection was compared in IC patients up to 90 days from the vaccine [23], but a comparison of the long-term duration of the specific immune response after vaccination or infection has not been described.

The objective of the current study was to compare the development and persistence of specific antibody and T-cell responses elicited by SARS-CoV-2 infection or the mRNA BNT162b2 vaccine.

## 2. Materials and Methods

### 2.1. Study Subjects

The immune response was evaluated in 37 post-COVID-19 patients enrolled after diagnosis of SARS-CoV-2 infection by nasal swab testing between March 2020 and December 2020 and 20 healthcare workers receiving a BNT162b2 vaccination between December 2020 and January 2021. The ethics committee approved the study protocols (P-20200046007, P-20210000232), and patients signed an informed consent.

Blood samples from 15 post-COVID-19 patients with pneumonia were evaluated in the convalescent phase of the infection (median: 59; range (45–90) days after infection), and 8 of them were analysed at a late time point (212; (186–400) days). In addition, 11 post-COVID-19 patients with mild symptoms were analysed at the early time point [48; (24–90) days], and 11 other post-COVID-19 patients with mild symptoms were analysed at late time point [193; (150–306) days].

Blood samples from 20 vaccinated subjects (with no previous SARS-CoV-2 infection) were collected three weeks (early time point) and six months (late time point) after two vaccine doses, and 10 of them were collected three weeks after the booster dose.

### 2.2. Antibody Assays

Anti-S IgG antibodies were determined using the SARS-CoV-2 Trimeric S IgG assay (Liaison, Diasorin, Saluggia, Italy), and results are given as BAU/mL (positive results > 33.8 BAU/mL). Additionally, the Nt Ab serum titre was determined as previously reported [24,25]. The results were considered positive if they were higher or equal to a 1:10 serum titre.

### 2.3. PBMC Isolation

Peripheral blood mononuclear cells (PBMCs) were isolated by standard density gradient centrifugation from heparin-treated blood using Lymphoprep (Sentinel Diagnostics, Milano, Italy). PBMCs were suspended in 10% dimethyl sulfoxide (DMSO) (Corning, NY, USA) and 90% heat-inactivated fetal bovine serum (FBS, Sigma, St. Louis, MO, USA) and stored in liquid nitrogen.

### 2.4. Detection of S-Specific T-Cell Proliferative Response

The S-specific proliferative response was determined as previously described [3]. Briefly, PBMCs (600,000/200 μL culture medium per well) were stimulated in triplicate with S and human actin peptide pools (15 mers, overlapping by 10 amino acids, Pepscan, Lelystad, The Netherlands) at a final concentration of 0.1 µg/mL for 7 days. After culturing, cells were washed and then stained with Live/Dead Fixable Violet Dye (Invitrogen, Waltham, MA, USA) and subsequently with CD3 PerCP 5.5 (BD Bioscience, Franklin Lakes, NJ, USA), CD4 APC-Cy7, CD8 FITC, CD25 PECy7 (all from BD Bioscience), CD278 (ICOS), and APC (Invitrogen). Finally, cells were washed and resuspended in PBS 1% paraformaldehyde.

A cell-proliferation index (CPI) was determined by subtracting the percentage of CD25^+^ICOS^+^, CD3^+^CD4^+^, or CD3^+^CD8^+^ detected in PBMC incubated with actin peptides from the percentage of CD25^+^ICOS^+^ T-cell subsets detected in PBMC incubated with S peptides. A CPI ≥1.5% was considered positive [3]. Flow cytometry analyses were performed with a FACS Canto II flow cytometer and BD DIVA software (BD Biosciences). A representative pseudocolor plot analysis is shown in Appendix A.

### 2.5. S-Specific Cytokine Production

The production of T_H_1 (IFNγ, TNFα, IL-2, MIP-1α, and MIP-1β) and T_H_2 (IL-4 and IL-5) cytokines and IL-10 was evaluated after 20 h stimulation of T cells with S protein or human actin peptides. Briefly, PBMCs were stimulated for 20 h with SARS-CoV-2-specific peptide pools from S and a peptide pool of human actin (1 µg/mL) in the presence of co-stimulator molecules CD28 and CD49d (BD Bioscience). Cells were seeded in 96-wells round-bottom plates at a density of 0.5–1 × 10^6^ cells/200 µL culture medium per well. The concentrations of cytokines and chemokines were measured in duplicate in cell culture supernatants using BioPlex Pro Human Cytokine Screening Panel (27-Plex #M500KCAF0Y, Bio-Rad, Hercules, CA, USA) as previously described [3]. All the results were analysed with BIO-PLEX Manager software version 6.2 (Bio-Rad).

### 2.6. Statistical Analysis

Statistical analyses were performed with GraphPad Prism 6. The Mann–Whitney U-test or Wilcoxon signed-rank test were applied for unpaired or paired comparison, respectively, while the Friedman and Kruskal–Wallis tests were applied for paired or unpaired multiple comparisons, respectively.

## 3. Results

### 3.1. Antibody Response Elicited by SARS-CoV-2 Infection or mRNA BNT162b2 Vaccination

Post-COVID-19 patients with pneumonia and vaccinated subjects developed higher levels of anti-S IgG response than post-COVID-19 patients with mild symptoms (Figure 1A). Anti-S IgG was higher in patients with pneumonia both at early (*p* < 0.01) and late (*p* < 0.001) time points than in patients with mild symptoms. In addition, vaccinated subjects developed higher levels of anti-S IgG (*p* < 0.001) than post-COVID-19 patients with mild symptoms at early and late time points, but no significant difference was observed between vaccinated subjects and post-COVID-19 patients with pneumonia. Patients with pneumonia showed sustained anti-S IgG levels from two months until seven months after infection, while anti-S IgG levels declined in mild-COVID-19 patients and vaccinated subjects after six months (*p* < 0.001) (Figure 1A).

A trend similar to that of anti-S IgG was observed for the Nt Abs. Post-COVID-19 patients with pneumonia and vaccinated subjects developed higher levels of Nt Abs than those of post-COVID-19 patients with mild symptoms (Figure 1B). The Nt Abs titre was higher in patients with pneumonia than in patients with mild symptoms at both the early and late time points (*p* < 0.05). Conversely, in vaccinated subjects, the levels of Nt Abs were higher than in post-COVID-19 patients with mild symptoms only at the early time point (*p* < 0.01), while the Nt titre declined in vaccinated subjects at the late time point (*p* < 0.001) to the range of mild COVID-19 patients.

### 3.2. Spike-Specific CD4^+^ and CD8^+^ T-Cell Proliferative Response after SARS-CoV-2 Infection or mRNA BNT162b2 Vaccination

Vaccinated subjects and post-COVID-19 patients showed comparable and sustained levels of specific CD4^+^ proliferative response at early and late time points (Figure 2A), with no significant decrease. However, in post-COVID-19 patients with pneumonia, we observed a significantly higher proliferative response than in mild COVID-19 patients at the early time point (*p* < 0.05). Moreover, it should be noted that while all patients with pneumonia showed detectable or borderline proliferative CD4^+^ T-cell response at both time points, 3/11 (27%) mild COVID-19 patients and 2/20 (10%) vaccinated subjects did not show a detectable response at the early time points. The frequency of no-responders at the late time point was 4/11 (36%) for mild COVID-19 patients and 5/20 (25%) for vaccinated subjects.

Instead, regarding the CD8^+^ T-cell proliferative response (Figure 2B) at the early time point, post-COVID-19 patients with pneumonia developed higher levels than patients with mild symptoms (*p* < 0.001) and vaccinated subjects (*p* < 0.05). At the late time point, the CD8^+^ T-cell proliferative response was still higher in post-COVID-19 patients with pneumonia than in vaccinated subjects (*p* < 0.05) (Figure 2B). Interestingly, only the CD4^+^ T-cell proliferative response was at the same high levels and persisted over time in both vaccinated subjects and post-COVID patients.

Regarding the cytokine profile of S-specific T-cells, post-COVID-19 patients with pneumonia and vaccinated subjects produced similar levels of T_H_1 cytokines, in particular IFNγ, IL-2, TNFα, and MIP-1β, which were significantly higher than in post-COVID-19 patients with mild symptoms (Figure 3A–E). On the other hand, the production of T_H_2 cytokines (IL-4 and IL-5) and IL-10 was poor or negligible in both post-COVID-19 patients and vaccinated subjects (Figure 3G,H).

### 3.3. Kinetics of Humoral and T-Cell Response before and after the Booster Dose of mRNA BNT162b2 Vaccine

The antibody and T-cell responses were investigated in 10 vaccinated subjects three weeks (early time point) and six months (late time point) after the second dose administration, and three weeks after the booster dose.

The levels of anti-S IgG and Nt Abs decreased significantly six months after the second vaccine with respect to levels observed at three weeks (*p* < 0.001; Figure 4A,B). However, after the booster dose, the prior higher levels of antibody response were restored (*p* < 0.001; Figure 4A,B).

In contrast, the CD4^+^ T-cell proliferative response persisted over time at levels similar to those observed three weeks after the second dose, with no significant decrease with time or increase after the booster dose (Figure 4C). Only one patient, with an absent CD4^+^ T-cell proliferative response after the second dose, developed this response after the booster. Instead, the CD8^+^ T-cell proliferative response decreased after the booster dose (Figure 4D).

## 4. Discussion

In this study, we compared the development and persistence of antibody and T-cell responses elicited by the mRNA BNT162b2 vaccine or SARS-CoV-2 infection. We analysed post-COVID-19 patients with pneumonia or mild symptoms around two and seven months after SARS-CoV-2 infection, while vaccinated subjects were analysed three weeks and six months after two vaccine doses, and three weeks after the booster dose.

Here, we showed that all post-COVID-19 patients with pneumonia and vaccinated subjects, as well as most patients with mild symptoms, developed a detectable and persistent antibody response (both anti-S IgG and Nt Abs) from the early to late time point (although the antibody levels decreased with time in mild-COVID-19 patients and vaccinated subjects).

Notably, post-COVID-19 patients with pneumonia and vaccinated subjects developed higher anti-S IgG and Nt Abs levels than post-COVID-19 patients with mild symptoms. We also observed a significant difference at the early time point in the proliferative CD4^+^ and CD8^+^ T-cell responses between post-COVID-19 patients with pneumonia or mild symptoms (but not between either the post-COVID-19 patient group or vaccinated subjects). However, conversely to antibody levels, the proliferative CD4^+^ T-cell response did not decrease significantly at the late time point in both post-COVID-19 and vaccinated subjects. Interestingly, both antibody and proliferative CD4^+^ T-cell responses developed at similar levels in post-COVID-19 subjects with pneumonia and vaccinated subjects. Conversely, at the early time point, CD8^+^ T-cell response was lower in vaccinated subjects than in post-COVID-19 patients with pneumonia, and decreased significantly at the late time point. We also investigated the cytokine production: both post-COVID-19 patients with pneumonia and vaccinated subjects developed similar levels of T_H_1 cytokines (and had a negligible T_H_2 response).Therefore, no major alteration in the T-cell cytokine profile was observed in the vaccine-elicited response with respect to natural infection.

We also observed the kinetics of humoral and T-cell response after an mRNA BNT162b2 booster dose. Anti-S IgG and Nt Ab titres declined in vaccinated subjects after 6 months, and were restored to the initial levels after the booster dose. On the contrary, the proliferative CD4^+^ T-cell response persisted over time and was not changed by the booster dose, while the proliferative CD8^+^ T-cell response decreased six months after vaccination, and was not restored after the booster dose but, surprisingly, decreased. We speculated that the booster dose elicited CD8^+^ T-cells with a rapid effector function but lacking proliferative potential, and therefore were not detectable using the assay adopted in our study.

Many studies investigated antibody and T-cell immunity in post-COVID-19 patients in the convalescent phase [1,2,3,4,5,6,7,8,9], as well as the long-term persistence of immunity [3,10,11,12,13,14]. Several studies, in agreement with our results, reported that IgG levels were higher in post-COVID-19 patients with severe symptoms [1,26,27], while another study reported no difference between mild and severe post-COVID-19 patients [28]. The immune response persisted until one year after SARS-CoV-2 infection, and it is likely that the immunological memory could persist for a longer time and may protect the immunocompetent subjects from SARS-CoV-2 secondary infections [14,29].

The B- and T-cell responses elicited by the mRNA BNT162b2 vaccine have a crucial role in the protection from SARS-CoV-2 infection and disease and in the establishment of a long-term memory response. In agreement with other studies [18,19,20,21,22], our data confirmed that the antibody levels were higher after the second dose of vaccine, but decreased after six months. We also observed that vaccinated subjects developed antibody levels similar to those of post-COVID-19 patients with pneumonia, but higher than those of mild-COVID-19 patients. Considering that most patients develop mild COVID-19 or asymptomatic SARS-CoV-2 infection, the higher antibody level observed in vaccinated subjects may be the reason for the higher protection from SARS-CoV-2 infection observed in subjects with vaccine-elicited rather than natural infection-elicited immunity [30], at least in the first months after vaccination. Although not significant, we also observed a trend towards a higher CD4^+^ T-cell proliferative response in vaccinated subjects than in mild-COVID-19 patients.

However, we observed that the antibody levels decreased six months after vaccination, as reported by other studies [18,19,20,21,22], and required a booster dose to be restored to the initial levels. On the contrary, the CD4^+^ (but not CD8^+^) T-cell proliferative response did not decline six months after the vaccine, but remained at the same high levels of the post-COVID-19 patients with pneumonia. This suggests that memory T-cells with proliferative potential also can survive when antibody levels decline, therefore providing potential protection in case of re-exposure to SARS-CoV-2. In addition, the T-cell response was found not to be affected by mutations in the S proteins causing escape from Nt Ab recognition [17]. The decline observed instead in the CD8^+^ T-cell response deserves further investigation.

The limitations of this study were the following: (i) the small sample size of post-COVID-19 patients and vaccinated subjects analysed; (ii) the cross-sectional analysis conducted in mild-COVID-19 patients; and (iii) the fact that post-COVD-19 and vaccinated patients were analysed at different time points because the two groups of subjects were previously enrolled in two different studies. However, we do not believe that this last issue may have influenced the results obtained, because we included samples in the study that were collected at roughly similar time points in the two groups. The added value was the comprehensive analysis of the long-term persistence of the different arms of the immune response elicited by the vaccine or natural SARS-CoV-2 infection. In conclusion, subjects vaccinated with the mRNA BNT162b2 vaccine develop antibody and T-cell responses similar to patients who experience COVID-19 pneumonia. The T-cell cytokine profile also overlapped, with a predominant T_H_1 response. On the contrary, levels of antibody and T-cells elicited in patients with mild-COVID-19 were lower, and this may explain the higher protection from SARS-CoV-2 infection conferred by the vaccine. Finally, while the antibody response required at least a booster dose to persist at sustained levels, the CD4^+^ T-cell response seemed to last longer. Future evaluations should analyse the long-term persistence of the immune response after the booster dose(s).

## Figures and Tables

**Figure 1 microorganisms-10-01250-f001:**
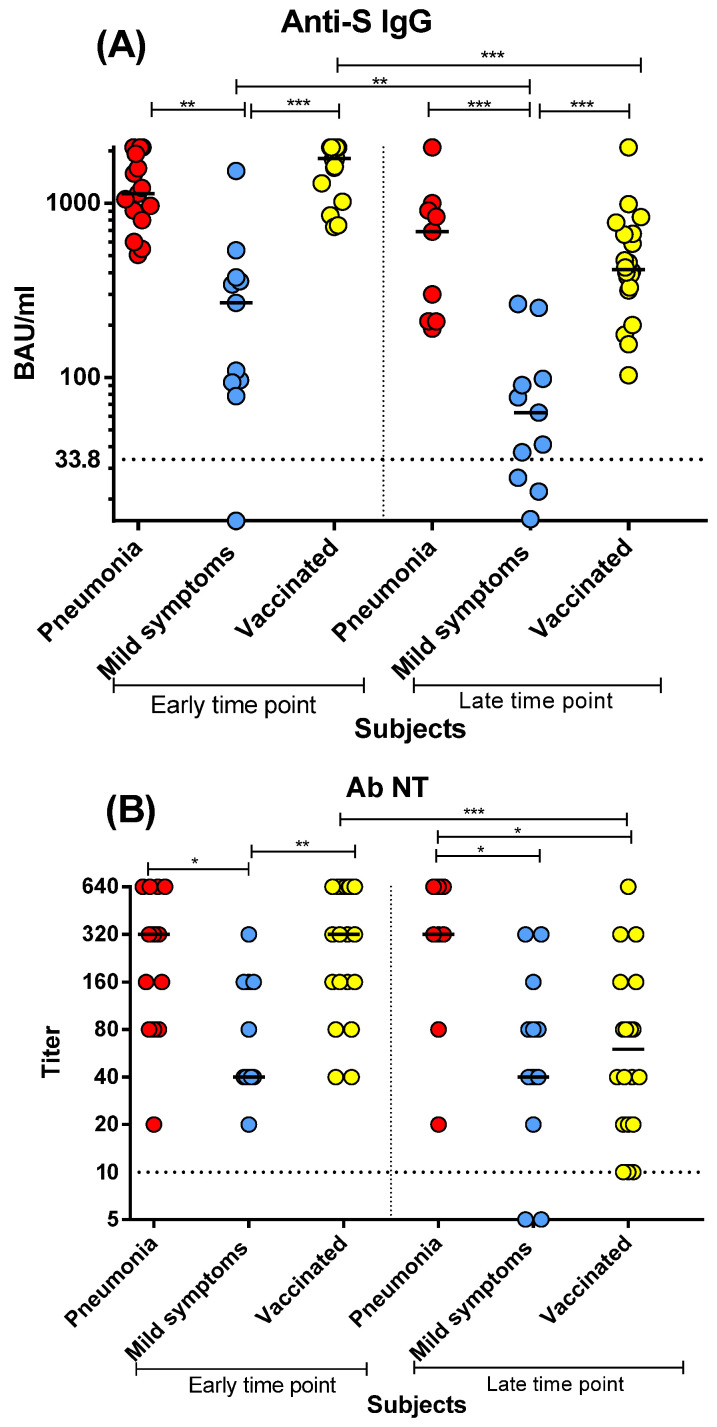
The spike (S)-specific antibody response was compared between post-COVID-19 patients with pneumonia or mild symptoms and vaccinated subjects at early and late time points: (**A**) anti-S IgG; (**B**) neutralising antibodies (Nt Ab). Early time point: patients with pneumonia, median 59 (range 45–90) days after infection; patients with mild symptoms, 48 (24–90) days after infection; vaccinated subjects three weeks after two vaccine doses. Late time point: patients with pneumonia, median 212 (range 186–400) days after infection; patients with mild symptoms, 193 (150–306) days after infection; vaccinated subjects six months after two vaccine doses. * *p* < 0.05, ** *p* < 0.01, *** *p* < 0.001.

**Figure 2 microorganisms-10-01250-f002:**
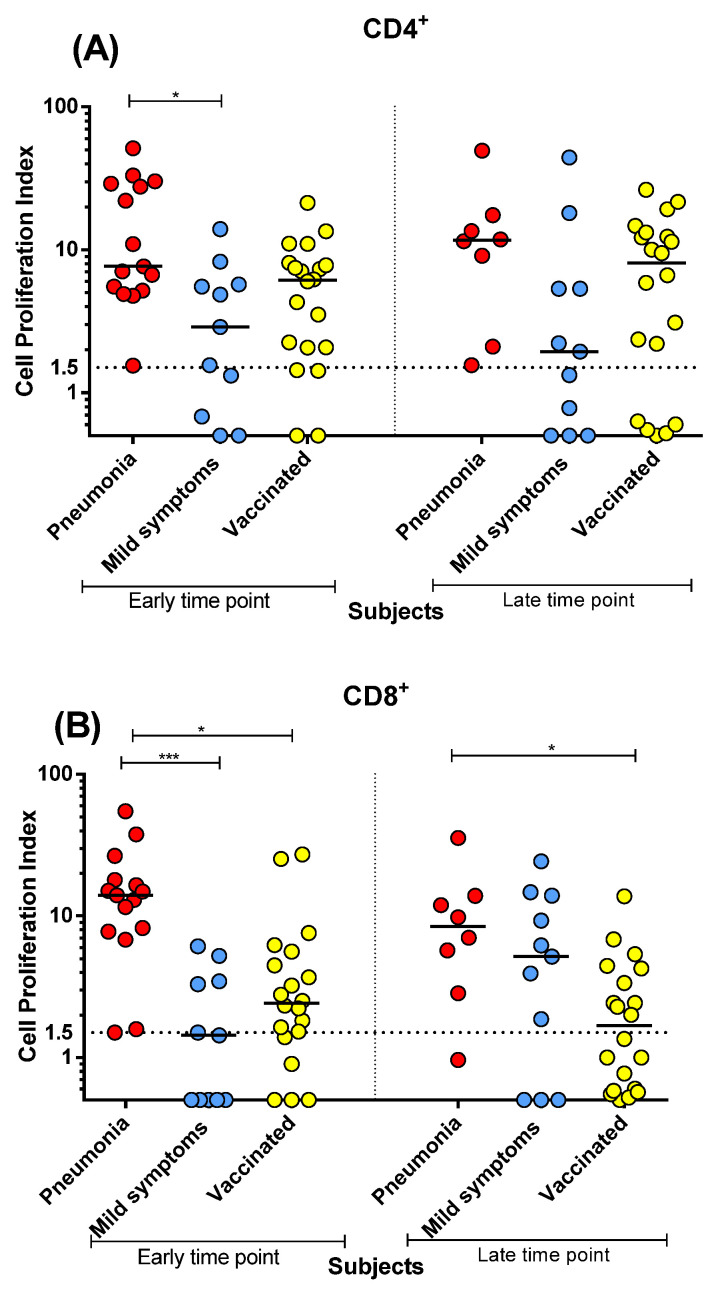
Spike (S)-specific T-cell response was compared between post-COVID-19 patients with pneumonia or mild symptoms and vaccinated subjects at early and late time points. (**A**) CD4^+^ T-cell proliferation index; (**B**) CD8^+^ T-cell proliferation index. Early time point: patients with pneumonia, median 59 (range 45–90) days after infection; patients with mild symptoms, 48 (24–90) days after infection; vaccinated subjects three weeks after two vaccine doses. Late time point: patients with pneumonia, median 212 (range 186–400) days after infection; patients with mild symptoms, 193 (150–306) days after infection; vaccinated subjects six months after two vaccine doses. * *p* < 0.05; *** *p* < 0.001.

**Figure 3 microorganisms-10-01250-f003:**
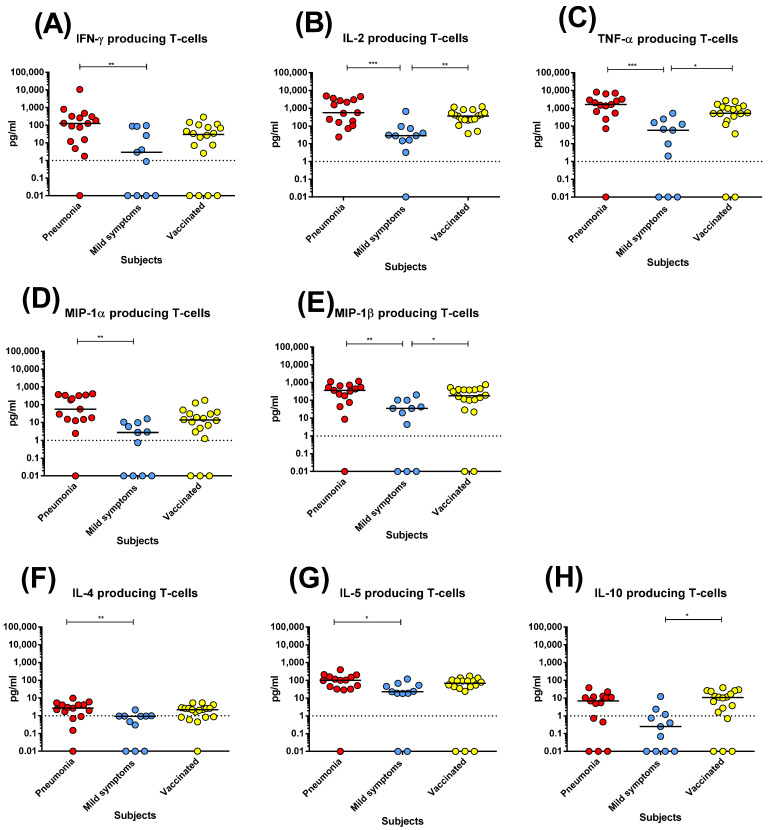
Cytokine production of T cells after stimulation with spike (S) peptide pools in post-COVID-19 patients with pneumonia or mild symptoms and vaccinated subjects at early time points: (**A**) IFN-γ; (**B**) IL-2; (**C**) TNF-α; (**D**) MIP-1α; (**E**) MIP-1β; (**F**) IL-4; (**G**) IL-5; (**H**) IL-10. Early time point: post-COVID-19 patients with pneumonia, median 59 (range 45–90) days after infection; patients with mild symptoms, 48 (24–90) days after infection; vaccinated subjects three weeks after two vaccine doses. * *p* < 0.05; ** *p* < 0.01; *** *p* < 0.001.

**Figure 4 microorganisms-10-01250-f004:**
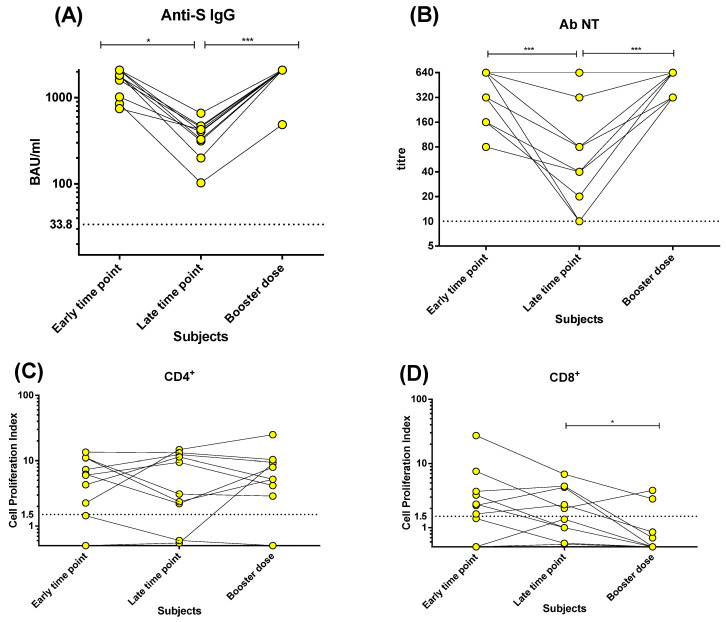
Spike (S)-specific antibody and T-cell responses were investigated in 10 vaccinated subjects receiving booster dose: (**A**) anti-S IgG; (**B**) neutralising antibodies (Nt Ab); (**C**) CD4^+^ T-cell proliferation index; (**D**) CD8^+^ T-cell proliferation index. Early time point: three weeks after two vaccine doses. Late time point: six months after two vaccine doses. Booster dose: three weeks after booster dose. * *p* < 0.05; *** *p* < 0.001.

## Data Availability

The data that support the findings of this study are available upon request from the corresponding author. The data are not publicly available due to privacy or ethical restrictions.

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
