# Peer review of "mRNA BNT162b Vaccine Elicited Higher Antibody and CD4+ T-Cell Responses than Patients with Mild COVID-19"

_microorganisms, 2022, doi:10.3390/microorganisms10061250_

Round 1

Reviewer 1 Report

In the manuscript "mRNA BNT162b Vaccine Elicited Higher Antibody and CD4+ T-cell Responses Than Patients with Mild COVID-19", the authors evaluated the development and persistence of antibody and T-cell responses elicited by mRNA BNT162b2 vaccine or SARS-CoV-2 infection. In 57 patients (37 post-COVID-19 and 20 vaccinated), anti-spike Ig-G and neutralizing antibodies were evaluated.

The results show that antibody levels are higher in patients given the vaccine than those with COVID-19.

The results are well explained, and all sections are written in detail.

Reviewer 2 Report

The objective of the current study was to compare the development and persistence of specific antibody and T-cell responses elicited in 37 post-COVID-19 patients and 20 vaccinated subjects (mRNA 59 BNT162b2 vaccine).

Comments:

1. The results of this work indicate that vaccinated subjects developed antibody levels similar to post-COVID-19 patients with pneumonia but higher than mild-COVID-19 patients. The antibody levels decreased six months after vaccination, unlike the CD4+ (but not CD8+) T-cell proliferative response that remains at the same high levels of the post-COVID-19 patients with pneumonia. Why CD8+ T-cell response decreased six months after vaccination, and was not restored after the booster dose but, on the contrary, decreased.  Could you explain this difference in a little more detail? This finding does not appear to have been found in other studies.    

2. Why have patients with COVID-19 infection and vaccinated patients been analyzed at different times, that is, in the first case around two and seven months after SARS-CoV-2 infection, while vaccinated subjects were analyzed three weeks and six months after two vaccine doses, and three weeks after the booster dose. This difference could have been avoided in the design of the study. Do you think that this fact may have influenced the results obtained?

Round 2

Reviewer 2 Report

The manuscript has been improved following the comments made, thank you very much